# Mixed Methods Lot Quality Assurance Sampling: A novel, rapid methodology to inform equity focused maternal health programming in rural Rajasthan, India

Aneel Singh Brar[1,2,3]*, Bethany L. Hedt-Gauthier[3,4], Lisa R. Hirschhorn[5]

1 Mata Jai Kaur Maternal and Child Health Centre, Sri Ganganagar, Rajasthan, India, 2 School of Anthropology and Museum Ethnography, University of Oxford, Oxford, Oxfordshire, United Kingdom, 3 Department of Global Health and Social Medicine, Harvard Medical School, Boston, Massachusetts, United States of America, 4 Department of Biostatistics, Harvard Chan School, Boston, Massachusetts, United States of America, 5 Medical Social Sciences, Feinberg School of Medicine, Northwestern University, Chicago, Illinois, United States of America

* aneel@matajaikaur.org

**Data Availability Statement:** All relevant data are within the paper and its Supporting Information files.

## Abstract

India has experienced a significant increase in facility-based delivery (FBD) coverage and reduction in maternal mortality. Nevertheless, India continues to have high levels of maternal health inequity. Improving equity requires data collection methods that can produce a better contextual understanding of how vulnerable populations access and interact with the health care system at a local level. While large population-level surveys are valuable, they are resource intensive and often lack the contextual specificity and timeliness to be useful for local health programming. Qualitative methods can be resource intensive and may lack generalizability. We describe an innovative mixed-methods application of Large Country-Lot Quality Assurance Sampling (LC-LQAS) that provides local coverage data and qualitative insights for both FBD and antenatal care (ANC) in a low-cost and timely manner that is useful for health care providers working in specific contexts. LC-LQAS is a version of LQAS that combines LQAS for local level classification with multistage cluster sampling to obtain precise regional or national coverage estimates. We integrated qualitative questions to uncover mothers' experiences accessing maternal health care in the rural district of Sri Ganganagar, Rajasthan, India. We interviewed 313 recently delivered, low-income women in 18 subdistricts. All respondents participated in both qualitative and quantitative components. All subdistricts were classified as having high FBD coverage with the upper threshold set at 85%, suggesting that improved coverage has extended to vulnerable women. However, only two subdistricts were classified as high ANC coverage with the upper threshold set at 40%. Qualitative data revealed a severe lack of agency among respondents and that household norms of care seeking influenced uptake of ANC and FBD. We additionally report on implementation outcomes (acceptability, feasibility, appropriateness, effectiveness, fidelity, and cost) and how study results informed the programs of a local health non-profit.

**Funding:** A.S.B. received funding from Harvard University and the Abundance Fund through the Harvard Medical School Department of Global Health and Social Medicine. The funders had no role in study design, data collection and analysis, decision to publish, or preparation of the manuscript.

**Competing interests:** The authors have declared that no competing interests exist.

## Introduction

In 2015, India accounted for 15% of all maternal deaths and 24% of all neonatal deaths globally [1, 2]. This high burden has persisted in the face of great progress in reducing maternal and neonatal mortality with reductions of 68.7% and 50.9%, respectively, over a period of fifteen years [1, 2]. Despite the progress in India and globally, the majority of maternal and child deaths continue to occur in lower income countries with the highest burden concentrated among the most vulnerable within these countries [3–5]. Ensuring that progress in reducing maternal and neonatal mortality includes improvements in equity is therefore both a moral imperative and a challenge requiring further innovation in approaches to both deliver and measure the impact of interventions targeting these populations. However, a major challenge to achieving health equity is a lack of understanding of the effectiveness of existing health policies and programs designed to target the poorest and most vulnerable populations.

India experiences high levels of maternal health inequity with poorer states and marginalized populations within those states experiencing worse health outcomes and lower levels of access to care [4, 6, 7]. To address inequity of access for pregnant women, in 2005 the Indian Government established a conditional cash-transfer program called Janani Suraksha Yojna (JSY or "safe motherhood scheme") within the National Rural Health Mission (NRHM) program that incentivized childbirth at government hospitals and birthing centers. Several assessments of the NRHM have shown improvements in facility-based delivery (FBD) and antenatal care (ANC) uptake, including among women with low literacy rates and from low-socioeconomic backgrounds [8–14].

Nevertheless, concerns about equity and the effectiveness of this strategy in reducing maternal and neonatal mortality persist [13–15]. While overall demand for FBD and ANC has increased, there has been a lack of concomitant investment in the quality and effectiveness of maternal health services, referral systems, and emergency care such as blood transfusions and Cesarean sections [4, 14, 16]. Additionally, the program's success in increasing access for the most vulnerable women, including those with high-risk pregnancies, has also been unclear [17–19].

Understanding the successes and challenges to implementation of equity-focused policies like JSY requires data collection methods that can produce a better contextual understanding of how vulnerable populations access and interact with the health care system at a local level, which in turn can help foster the design, adaptation and delivery of effective, "people-centered" care [20, 21]. While national and subnational data on health indicators are available in most low- and middle-income countries, these quantitative population-level surveys do not capture the lived-experience of care-seeking nor how and why a health care system is or is not working for specific populations in more local settings [22]. These population surveys are also costly, are only repeated at long intervals, and can lack adequate subnational granularity limiting their utility for measuring and improving local outcomes through contextually responsive health programs for underserved populations. On the other hand, more traditional qualitative or ethnographic methods can be resource intensive, difficult to repeat at the frequency needed to inform change and, while providing depth, may lack generalizability across a local population targeted for a health intervention.

Lot Quality Assurance Sampling (LQAS), a binary classification tool to identify poorly performing groups or lots, has gained in popularity in global health applications as a way to quantitatively assess outcomes at the local level and provide low-cost, timely, and relevant information to program managers [23–33]. LQAS classifies a given supervision area as either 'high' or 'low' according to a pre-specified coverage target and maximum allowable misclassification errors. Efforts to improve the validity and applicability of LQAS have included

incorporating Bayesian statistics and cluster sampling techniques [34–37]. No previous application, however, has integrated qualitative data gathering into the LQAS study design to provide an additional layer of patient-reported information.

We developed and implemented an innovative mixed-methods application of Large Country-Lot Quality Assurance Sampling (LC-LQAS). LC-LQAS is a version of LQAS that combines LQAS for local level classification with multistage cluster sampling to obtain precise regional or national coverage estimates [38]. We applied mixed-methods LC-LQAS to rapidly obtain estimates of FBD and ANC coverage and to uncover mothers' experiences accessing maternal and reproductive health care in the rural district of Sri Ganganagar, Rajasthan, India. We also captured selected implementation outcomes as defined by Proctor et al. [39]. The objective was to use the results of our study to inform the needed adaptations of the maternal health programs of a local non-profit, the Mata Jai Kaur Maternal and Child Health Centre (MJK), and to understand the potential value and implementation of this measurement approach for driving data-informed improvement at local levels. We describe and report on selected implementation outcomes, including acceptability, feasibility, appropriateness, effectiveness, fidelity, and costs of mixed-methods LC-LQAS with reference to survey results for FBD and ANC and how these results informed MJK's program adaptation.

## Methods

### Setting and context

MJK is a non-profit organization that provides antenatal care and safe childbirth services free of cost to low-income and vulnerable women in the rural district of Sri Ganganagar, the northernmost district in the state of Rajasthan (Fig 1). MJK's catchment area includes three of Sri Ganganagar's nine *tehsils*, or sub-districts: Padampur, Raisinghnagar, and Karanpur (Fig 2) with a population of 359,746 [40]. The *Tehsils* are divided into 118 *panchayats*, groupings of five to eleven villages. Since 2009, MJK has provided ANC to over 18,000 outpatient women, the majority of whom come from low-income families from remote villages often with complicated pregnancies [41].

Sri Ganganagar has experienced a marked improvement in maternal mortality (343 deaths per 100,000 live births in 2010 [42] to 191 in 2013 [9]) along with increasing FBD (32.2% in 2005 [43] to 88.8% in 2016 [44]) and JSY coverage (26.8% in 2007 [45] to 60.1% in 2015 [44]). Despite this progress, MJK continues to receive patients from low-income families who did not receive JSY or encountered other barriers to accessing government health facilities for ANC or childbirth. These include remoteness from the facility and socio-cultural barriers; a belief that private or non-governmental care is of better quality than government care, especially for those with complicated pregnancies or who require surgery; and an unwillingness to return to government care based on previous poor experiences. The gap between the district-wide progress in coverage and the needs and experiences of MJK patients motivated the development of mixed-methods LC-LQAS. Our objective was to gather the evidence required to drive MJK's program adaptation and to inform government health policy to improve equity in access and care. LQAS could provide MJK managers with a rapid assessment of whether their target population was receiving FBD and ANC. Integrating cluster sampling and qualitative methods adds value by providing regional point estimates that can be compared to national-level survey data and further contextual details to help inform program design.

### The mixed-methods LC-LQAS study design and analysis

Our mixed-methods LC-LQAS had two distinct components. First, we used LC-LQAS design principles to identify a cross-sectional sample to collect quantitative indicators. Second, we

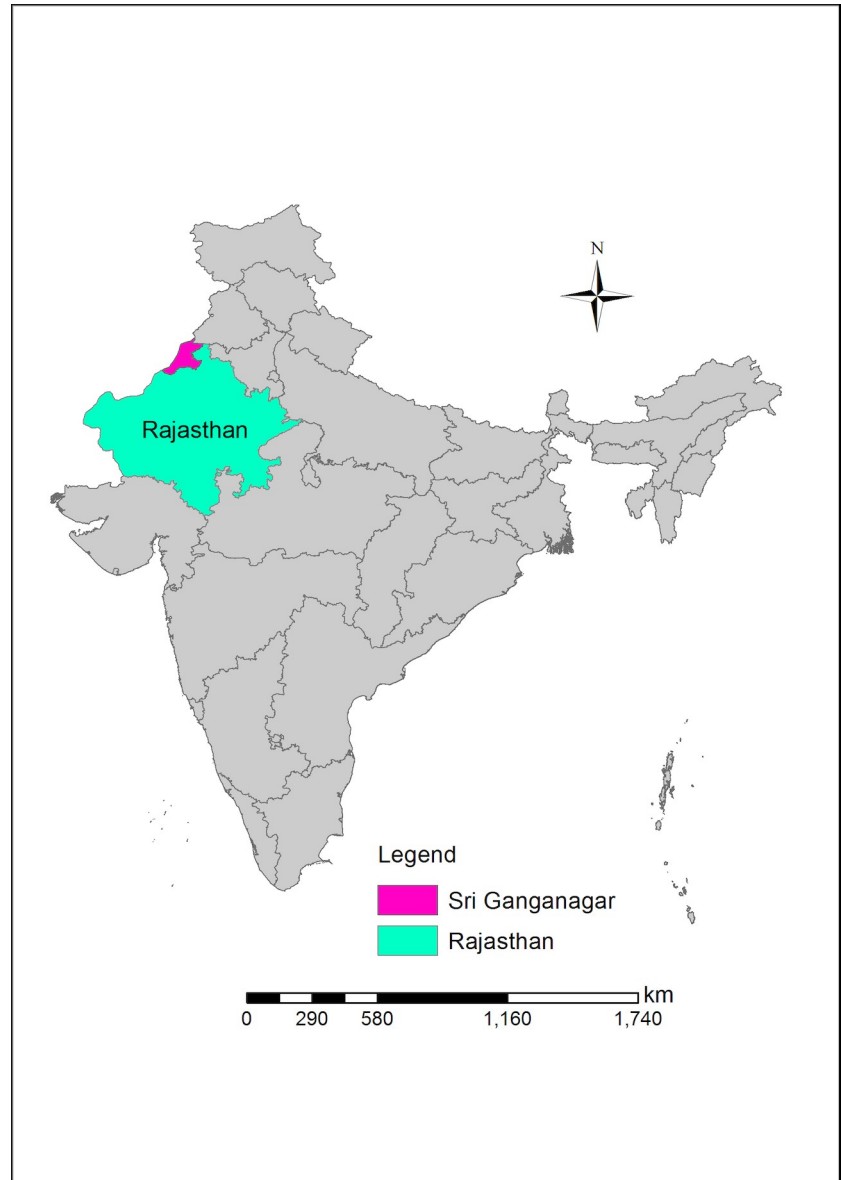

**Fig 1. Map of India with Rajasthan and the district of Sri Ganganagar highlighted.**

followed the participants' responses with a set of qualitative questions tailored to their responses to the quantitative portion of the interview. These are described in more detail below.

**Large-Country Lot Quality Assurance Sampling: Indicators, parameters and sample size.** The primary indicators for the LC-LQAS survey were: 1) "Full" ANC; defined by Indian population surveys at the time of this study as receiving at least three ANC checkups, at least one tetanus vaccination, and 100+ iron folic acid tablets or equivalent syrup received and consumed [9] and 2) FBD defined as delivery in a health care facility, regardless of whether a public or private-sector facility. Our choices for the LQAS parameters for ANC and FBD are based on the most recently available population level surveys at the time of the study, potential influence of JSY, and the experience of our target population (S1 File). For full ANC coverage we set a lower threshold of 10% and an upper threshold of 40%. For FBD we set the lower and upper thresholds at 55% and 85%, respectively. Maximum allowable misclassification errors at

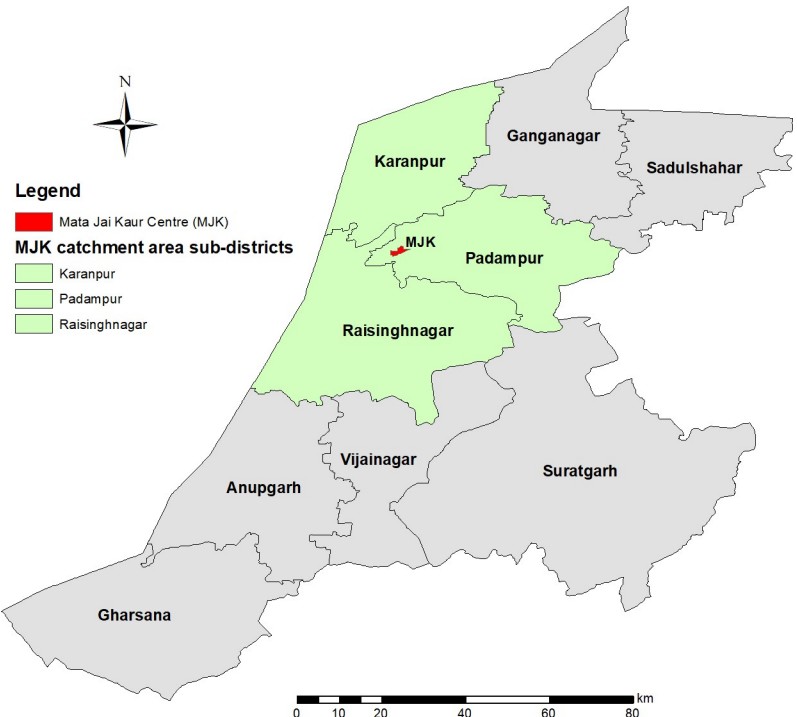

**Fig 2. Sub-districts (*tehsils*) and MJK catchment area in Sri Ganganagar.**

each threshold (typically called α and β errors) for both indicators were set at 10%. These LQAS parameters resulted in a per-cluster sample size of 15 for ANC and 16 for FBD; our study used the largest of the two (m = 16), with corresponding decision rules of 4 and 12 for ANC and FBD respectively.

To obtain regional point estimates with confidence intervals restricted to +/- 10%, we used two-stage cluster sampling with procedures described by Hedt et al. [38]. In the first stage, simple random sampling was used to select *panchayats* (clusters). In the second stage, a random sample of minimum 16 women were selected within each chosen *panchayat*. We assumed an intracluster correlation coefficient of 0.10. Given the most recent population size estimates for women aged 18–49 in Sri Ganganagar, the required sample size for randomly sampled clusters (*panchayats*) was 18, with 16 women per cluster, resulting in a total of 288 women. These calculations were performed on the basis of the number of women aged 18–49 years in each village from the 2011 Indian census (data on ages 15–17 years were not available) (S2 File).

**Study population and sampling procedures.**   Eligible respondents included low-income women between the ages of 15 and 49 years. Low-income was defined as women residing in *kachha* (low-quality, mudbrick construction) or semi-*pucca* (50% or less brick or cement construction) houses, which are materials associated with poverty in India [46]. To capture pregnancy and delivery experiences in the study area, respondents also had to have lived in the household for at least two-thirds of their pregnancy (approximately 6 months for a full-term pregnancy), delivered a live infant within the last two years (since August 2013), be present when the enumerator visited the house, and provide consent, or assent in the case of minors, to participate in the study.

Since each *panchayat* consists of multiple villages of differing populations, the number of women to be sampled from each *panchayat* to meet the total sample size of 16 was distributed

proportionately amongst the constitutive villages with probability proportionate to size based on the number of women aged 18–49 from the 2011 Indian Census.

Upon visiting a village, the local Accredited Social Health Activists (ASHAs) or *angadwadis* (local, female government health workers responsible for maintaining complete birth registries) were consulted to create a list of women in their registries between the ages of 15 and 49 years who had delivered a live infant within the last 2 years. Using a random number generator, a random sample was generated to determine the order of initial home visits. This process of randomization was repeated as needed to generate additional houses if an inadequate number of eligible participants were identified from the first list. ASHAs and *angadwadis* were then recruited on a volunteer basis to help guide the survey team to the sampled houses. If a village was large enough to have multiple ASHAs and birth registries, the number of women to be sampled for that village was divided evenly between the ASHA's catchment areas and separate, randomly generated lists developed for each area.

When reaching a house, the building material was inspected visually from both the outside and, if given access, inside from the entry way. If eligible based on house materials, permission for access to the house was obtained to determine full eligibility, including consent. Women from the second and subsequent lists were substituted sequentially as needed until the quota of respondents for that village was achieved. If this process did not yield enough participants for a given village (for example, due to too few responses or mud-walled houses), the numbers were added to the targeted number for the next village. One *panchayat* was surveyed per day, with an extra day reserved for follow-up to ensure adequate sample sizes were obtained.

**Data collection.**   Quantitative data collected included general demographic information, and for the most recent pregnancy, ANC attendance and receipt of and adherence to folate and iron supplementation, receipt of the JSY subsidy, and whether the delivery was facility-based. Demographic variables included household type (*kachha* or semi-*pucca*), education level, caste designation (Scheduled Caste, Scheduled Tribe, or Other Backwards Class) and Below Poverty Line (BPL) status. These caste and tribe designations are official statuses recognized in the Indian Constitution for groups of people who are economically or socially disadvantaged. BPL households are assessed as economically disadvantaged by local government agencies and provided with a BPL card which entitles them to certain welfare schemes.

**Integrated qualitative questions.**   Respondents were asked qualitative questions about the facilitators and barriers to care depending on their reported receipt of ANC and FBD. For example, depending on whether a respondent answered "yes" or "no" to the question "did you have a facility-based delivery?," a series of questions were triggered to elicit qualitative responses specific to that outcomes. A "no" response would be followed by:

- Tell me the reasons you did not go to a clinic or health facility for your last delivery?

- Did you want to deliver in a health facility? If yes, what prevented you from delivering in a health care facility? If no, why did you not want to deliver in a health facility?

- Is there anything that could have motivated you to deliver at a health facility?

   A "yes" response was followed by:

- What were the reasons you delivered in a health facility?

- Explain how this helped you get to a hospital or clinic for delivery?

- Did somebody make it difficult for you to deliver at a facility?

   Regardless of FBD outcome, all women were asked the question "can you tell me of the story of your delivery day?"

An analogous series of questions related to ANC followed the "yes" or "no" responses to the question "did you receive antenatal care at a clinic or health facility?" This was followed by the open-ended question, "is there anything else you would like to tell me about your experience *during* pregnancy?"

**Analysis of the mixed-methods LC-LQAS data.**   Using standard LQAS methods, *panchayats* were classified as being high or low coverage based on whether the number of sampled women in the *panchayat* that reported FBD or full ANC was greater than the respective decision rule for that indicator [38, 47]. Descriptive statistics including demographics and receipt of JSY subsidy and catchment area coverage proportions with 95% confidence intervals were determined using Stata version 13.1. The complex survey design was accounted for in the analysis using the *svy* set of commands within Stata. Qualitative interviews were transcribed and translated into English and coded by AB using an inductive, content-focused approach. Higher-level themes were reviewed by LH, BHG. Coding was stopped when saturation was recognized. Similar codes were grouped into broader categories that characterized participant experiences, facilitators and barriers related to care-seeking behavior or health care provision. Qualitative data were managed using Dedoose version 7.0.25.

## Assessment of the mixed-methods LC-LQAS

We adapted the implementation outcomes from Proctor et al. [39] to assess the success and challenges of our mixed methods survey approach. For this study, we defined the outcomes as follows:

- Acceptability: 1) The acceptance of the sampling procedures and surveys among respondents and their household members, community health workers (CHW), and local health bureaucrats; and 2) The acceptance of respondents to the integrated qualitative questions. To assess acceptability, we collected refusal information from eligible households that were approached for the survey. We recognized the importance of collecting refusal information after completing the first four panchayats, and so only collected and included this information from the subsequent 14.

- Feasibility: The extent to which the methodology was successfully implemented within MJK's setting, including ability to conduct the surveys, time required for implementation and the ability to get CHWs and respondents to participate.

- Appropriateness: The perceived fit and relevance of mixed-methods LC-LQAS to achieve our study objectives within the given context and resource constraints. Appropriateness is from the perspective of program implementers, evaluators, researchers, or local stakeholders interested in understanding and improving implementation or policy.

- Effectiveness: Whether the use of mixed-methods LQAS produced data resulting in a better understanding of FBD and ANC coverage and access among vulnerable women, and if this evidence informed MJK's programs.

- Fidelity: The degree to which the method was implemented and data collected as intended in a real-world context [39, p.70].

- Cost: The cost in USD of implementing the mixed-methods LQAS.

## Ethics

The study was approved by the IRB at Harvard University Faculty of Medicine and the Bio-Medical Ethics Committee in New Delhi. Oral informed consent was obtained prior to the start of the survey. Oral consent was used for this population because of high levels of illiteracy,

which is estimated at 67.2% for women in rural Sri Ganganagar, Rajasthan [40]. If the participant was between the ages of 15 and 18, oral informed consent was obtained from their parent or guardian in addition to informed assent from the participant herself. Informed consent and assent were confirmed and documented on a form that was signed by the researcher.

## Results

### Results of the mixed methods LC-LQAS study

**Quantitative results.** All of the 18 sampled *panchayats* were classified as achieving the coverage target of 85% for FBD. The point estimate for FBD coverage was 90.8% (95%CI [86.0%, 94.0%]). A large majority (84.4% (95%CI [78.4, 89.0])) of respondents received the JSY subsidy. For full ANC, only two of 18 *panchayats* were classified as achieving the coverage target of 40% with a corresponding point estimate of 13.1% (95% CI [9.3, 18.3]) (see Table 1). The major gaps for full ANC included receiving a full supply of iron and folate (31.4% (95% CI [25.2–38.3])) and adherence to taking the full course if received (15.7% (95% CI [11.6–21.0])).

**Qualitative results.** Three themes and nine sub-themes emerged from content analysis of the qualitative data (Table 2). The level of family support was the most significant barrier or facilitator to care for both ANC and FBD, reflecting a lack of agency or decision-making power for the mother. Regardless of a respondent's intention to seek care, knowledge of its importance, or the availability of funds or transportation, accessing care depended on a whether the household had developed a norm of care seeking.

### Assessment of the mixed-methods LC-LQAS

**Acceptability.** In all 18 *panchayats*, we approached a total of 456 eligible households from which 313 (68.6%) women were surveyed, all of whom participated in both qualitative and

**Table 1. LQAS results for indicator "facility-based delivery" and "full antenatal care" with sample sizes, decision rule, and whether target coverage was reached (yes —✓; no—x) (n = 309).**

| *Panchayat* (supervision area) | Sample size | Received a facility-based delivery (Target: ≥85%) | | Received a Full antenatal care (Target: ≥40%) | |
|---|---|---|---|---|---|
| | **n** | **d** | **Target reached** | **d** | **Target reached** |
| 2 X | 18 | 14 | ✓ | 5 | x |
| 36 H | 16 | 12 | ✓ | 4 | x |
| 42 H | 16 | 12 | ✓ | 4 | x |
| 43 GG | 18 | 14 | ✓ | 5 | x |
| 50 F | 16 | 12 | ✓ | 4 | x |
| 1 PS | 16 | 12 | ✓ | 4 | x |
| 3 EEA | 18 | 14 | ✓ | 5 | x |
| 34 LNP | 18 | 14 | ✓ | 5 | x |
| 4 DD | 16 | 12 | ✓ | 4 | x |
| 6 RB | 19 | 14 | ✓ | 5 | x |
| 75 LNP | 18 | 14 | ✓ | 5 | x |
| Bingh Bayala | 16 | 12 | ✓ | 4 | x |
| 15 PTD-A | 17 | 13 | ✓ | 4 | x |
| 2 IWM | 17 | 13 | ✓ | 4 | x |
| 22 PTD-B | 18 | 14 | ✓ | 5 | x |
| 23 RB | 16 | 12 | ✓ | 4 | ✓ |
| 75 NP | 17 | 13 | ✓ | 4 | x |
| Bhompura | 19 | 14 | ✓ | 5 | ✓ |

**Table 2. Themes and sub-themes from qualitative data.**

| Themes and sub-themes | Sub-themes |
| --- | --- |
| 1. Lack of agency | • Household norm of care seeking |
| | • Supply and demand-induced behavior change |
| | • Potential agency |
| | • Burden of domestic responsibility |
| | • Delivery at *Mayka* (natal home) |
| 2. Hidden violence | • Uncertain care journeys and referrals |
| | • Calculating costs versus risks |
| | • Amassing funds |
| 3. Women's worth and gender-preference | • Assumed potential of having a son |

quantitative components of the study, providing interviews of approximately 15–20 minutes in length. Five of these respondents (1.6%) refused audio recording. We captured specific refusal information from 14 of 18 *panchayats* in which we approached 415 eligible households with 242 (48.3%) having a woman who delivered in the previous two years of the survey at home. Only three of the 242 (1%) refused to participate (1 language barrier, 1 refusal by head of household and 1 did not give consent).

ASHAs and *angadwadis* in every village (80 in 79 villages) responded positively to our requests for assistance, although the nature of their assistance varied from only providing names to physically helping us locate households and respondents. Gaining local governmental approval and buy-in to conduct the study was crucial to facilitating the acceptance of community health workers in allowing us to access birth registers and helping to locate households. In one instance we were initially blocked from access to a village's registers by a resident Auxiliary Nurse Midwife, the ASHA supervisor, until we produced the letter of support from the local Chief Medical and Health Officer. Although we were never explicitly requested to show this letter of support in any other village, we made it a routine process after this encounter.

## Feasibility

Implementing the mixed-methods LC-LQAS was feasible. We approached 456 eligible households in 18 *panchayats* consisting of 151 villages. The survey was completed over 19 days between September 16, 2015 and October 10, 2015 with all targeted *panchayats* reached, sample sizes achieved, and data collected according to the original or adapted protocols (see Fidelity). Data collection required a team of seven individuals and two cars covering distances of 100 to 150 kilometers a day in order to survey an average of over 4 villages a day. At this pace, each survey day was intensive, requiring between 8 to 12 hours of work in the field and up to 2 hours of post-fieldwork data processing and debrief during which notes on each household, respondent, and ASHA encounter were compiled, difficulties reviewed, and a strategy for the next day's survey developed. Two teams led by two female research assistants simultaneously surveyed different villages within a *panchayat*. Reaching villages and obtaining birth lists required approximately 2 to 3 hours depending on several variables including distance and ASHA availability and the state and organization of her records. Approximately two households were physically approached or sought out for every eligible household found. The most common reason for an excluded household was that the woman was not home, often because she had moved to her natal village (*mayka*)—a common local practice after the birth of the first child—was away for work or was attending a religious or family function. The most

common reasons that a household was deemed ineligible was its status as a *pucca* or wealthy house (18.15%); the respondent not having lived in the village during their pregnancy (0.97%); and miscarriage, stillbirth, or infant death of the most recent child (0.77%). Completion of the survey including the qualitative questions required between 20 to 40 minutes.

Feasibility required careful planning and preparation and intensive commitment from the survey team, including drivers, given long distances, variable road conditions, and long working hours. The challenges were greater for more remote *panchayats* which tended to also have larger distances and poorer road conditions between villages. Feasibility was increased in several ways. First, substantial pre-deployment field testing and training reduced the time required to conduct the survey achieving a reasonable balance between speed and the quality of data obtained. The research assistants' fluency in multiple local dialects and their awareness of cultural, religious, and social differences between households helped in terms of accessing households and respondents and increasing the ability to complete the survey. Lastly, the strategy of involving community health workers increased feasibility by reducing the time and resources needed to sample *panchayats* and find the selected respondents.

**Appropriateness.**    The LC-LQAS results obtained suggest that the methodology was an appropriate tool to determine FBD, ANC, and JSY coverages for the target population. Further, the integrated qualitative questions added depth to survey responses in a way that was targeted and responsive to the experiences of the mother.

The use of *kachha* and semi-*pucca* houses ensured that we successfully targeted women with lower socioeconomic status relative to other households in Rajasthan (Table 3). Two-thirds (61.6%) of respondents had a primary (up to grade 5) or lower level of education, with 23.1% having never attended school. The majority (62.1%) had Scheduled Caste status, an indicator of lower social and economic opportunity, with an average respondent's self-reported monthly household income of $17 USD per capita. Comparatively, Scheduled Castes constitute only 18.5% of the rural Rajasthani population and the average household income in Rajasthan is $83 per capita [48–50]. The women surveyed had a mean age of 23.6 (95%CI [23.2, 24.0]) years.

## Effectiveness

The survey was effective in generating LQAS classification and overall coverage estimates that allowed MJK to see whether trends in national and district level surveys applied to vulnerable women in its context and to identify local barriers and facilitators to access.

Our quantitative results suggest that NRHM and JSY are having a positive influence on access including among low-income households. Our coverage estimates for FBD (90.8%) and JSY (84.4%) for low-income, rural women in MJK's catchment area equaled or surpassed the respective coverages for all women, regardless of income, in the most recent district level health survey, the National Family Health Survey-4 (NFHS-4) [46] (Table 4). These results suggest that the cash incentive continues to facilitate the targeted care seeking behavior.

Our coverage estimate for full ANC (13.1% (95% CI [9.3, 18.3])) for low-income women was low and unchanged from similar data from the district from over a decade earlier in the 2007 District Level Household Survey (13.0%), which utilized the same definition for full ANC, as well as the most recent NFHS-4 (2015–16) for rural Sri Ganganagar (14.2%), which has a similar definition of full ANC except for having four rather than three check-ups [44, 45]. This unchanging coverage suggests that the behavior change in accessing facility-based delivery was not associated with improvements in full ANC including the minimum number of visits, tetanus vaccination, and receiving and taking of iron and folate supplementation. While full ANC rates were low, we did see higher coverage of components similar to or even

**Table 3. Survey population characteristics (n = 313).**

| Characteristic | Weighted n | % | 95% CI |
|---|---|---|---|
| **Household type** | | | |
| Kachha | 97 | 30.8 | (23.3, 39.6) |
| Semi-pucca | 216 | 69.2 | (60.4, 76.7) |
| **Education category**[a] | | | |
| Never attended school | 72 | 23.1 | (18.0, 29.2) |
| Did not complete primary | 22 | 6.9 | (4.9, 9.7) |
| Completed primary (5th) | 99 | 31.6 | (24.5, 39.7) |
| Completed upper primary (8th) | 58 | 18.4 | (12.2, 26.8) |
| Completed secondary (10th) | 35 | 11.3 | (8.1, 15.6) |
| Completed plus 2 or higher | 27 | 8.65 | (6.4, 11.5) |
| **Caste designation** | | | |
| None | 8 | 2.5 | (1.2, 5.1) |
| Scheduled Caste | 194 | 62.1 | (48.1, 74.4) |
| Other Backwards Class | 107 | 34.2 | (23.3, 47.0) |
| Unsure | 4 | 1.19 | (0.17, 8.0) |
| **Married** | | | |
| Married | 312 | 99.7 | (97.8, 1.0) |
| **Age (n = 274)** | | | |
| Mean years, SD, 95%CI | 23.6 | 3.09 | (23.2, 24.0) |
| **Below Poverty Line** | | | |
| Has BPL card | 60 | 19.3 | (14.1, 25.9) |
| **Owns agricultural land (n = 312)** | | | |
| Owns land | 158 | 50.6 | (40.5, 60.7) |
| **Monthly household income (n = 139)**[b] | | | |
| Mean USD, SD, 95%CI | 97.50 | 70.3 | (80.3, 114.7) |
| **Income per house member (n = 139)**[b] | | | |
| Mean USD, SD, 95%CI | 16.80 | 12.2 | (14.7, 18.9) |

[a]Rajasthani school system categories: primary (grades 1–5); upper primary (6–8); secondary (9–10); plus 1 and plus 2 (11–12); college, university.
[b]Conversion used is 1 INR = 0.015 USD.

higher than the most recent district level 2012–13 Annual Health Survey [9]. For example, 99.0% (95% CI [97.1, 99.6]) of women we surveyed reported receiving at least 1 tetanus vaccine compared to 92.8% in 2013 and 86.5% (95% CI [82.0, 90.1]) reporting 3 or more ANC checkups

**Table 4. Facility based delivery (%) and JSY beneficiaries (%) in rural Sri Ganganagar, Rajasthan according to different surveys.**

| | Survey and sample year | | | | |
|---|---|---|---|---|---|
| | NFHS-3 (2005) | DLHS-3 (2007) | JSY-E (2009) | AHS (2012) | NFHS-4 (2015) |
| Facility-based delivery | 23.3 | 40.7 | 59.1 | 84.2 | 91.9 |
| JSY beneficiary | - | 26.8 | 52.0 | 63.8 | 60.1 |

Eligible women for JSY benefit in Rajasthan are all pregnant women aged 19 and older. The four surveys from which these coverage estimates are taken are: 1. National family health survey (NFHS-3), 2005–06 [51]; 2. District Level Household and Facility Survey (DLHS-3), 2007–08 [45]; 3. Concurrent Assessment of Janani Suraksha Yojana (JSY) in Selected States: Bihar, Madhya Pradesh Orrisa, Rajasthan, Uttar Pradesh [52]; 4. Annual Health Survey: 2012–2013 [9]; and 5. National family health survey (NFHS-4), 2015–16 [44].

Table 5. LQAS sample size (n) and point estimates for ANC component indicators.

| ANC Component Indicator | n | Proportion (%) | 95% CI |
|---|---|---|---|
| At least 1 tetanus vaccine received | 312 | 99.0 | (97.1, 99.6) |
| Received 3+ ANC check-ups | 313 | 86.5 | (82.0, 90.1) |
| Received any IFA supplements | 312 | 88.9 | (84.6, 92.1) |
| Received adequate supply of IFA (100 pills or equivalent syrup) | 309 | 31.4 | (25.2, 38.3) |
| 100+ days of IFA received and consumed | 312 | 15.7 | (11.6, 21.0) |
| Received "Full" ANC | 310 | 13.1 | (9.3, 18.3) |

compared with 63.4% (no CI intervals provided in the 2012–13 Annual Health Survey report) (Table 5). These results are particularly hopeful in that we targeted women who were of low income, a population for whom inequity in other care areas has been reported [53]. However, similar rates and change were not seen in some components including self-reported iron and folate supplementation. Only 31.4% (95% CI [25.2, 38.3]) of women received an adequate supply of iron and folic acid supplementation and even fewer women reported having consumed at least 100 days of iron and folate acid supplementation (15.7% (95% CI [11.6, 21.0]) in our survey compared with 21.6% versus in the 2012–13 Annual Health Survey [9]. Barriers to adherence identified from the qualitative interviews included the taste of the supplement and a lack of belief that it was needed, though there may be other structural or access issues.

Our LC-LQAS data confirmed that previous larger population level survey results applied to low-income women in our local area and, in the case of FBD and JSY, suggested that the trend in increasing coverage has continued. While we found high coverage for FBD and ANC visits (although not all components), our quantitative results do not explain the continuing flow of low-income and vulnerable women seeking services from MJK, nor resolves persistent concerns about inequitable access to maternal health care and health outcomes in Rajasthan and India more broadly [5, 12, 14, 15, 17–19]. India remains the 6th most inequitable country in the world for maternal care with a coverage gap of 48.2% and 28.7%, for ANC visits and having a skilled attendant at delivery, respectively, between the richest and poorest quintiles [5].

MJK was founded with the objective of addressing inequitable access by providing FBD and ANC services free-of cost in a remote, village-based setting. Over 96% of MJK's patients are from low-income families and many require specialized care for complicated pregnancies that are perceived to not be available or of sufficient quality at government centers [41]. MJK's patient profile suggests that perceived or actual quality of care at government centers, the costs of private medical care, poverty, and remoteness are barriers to care and factors that may contribute to the continued flow of low-income patients to MJK. Our qualitative results identified additional factors related to the respondents' household and family practices that MJK could address to further its mission of improving equitable access.

One of the most important barriers and potential facilitators to health care access was the woman's decision-making power within the household, highlighting the profound lack of agency the mothers in our study had regardless of whether they were or were not successful in accessing ANC, FBD, or a JSY cash transfer. Of the few respondents who delivered at home, many did so after a previous negative experience with childbirth at a hospital with several mentioning comfort and the lack of a female practitioner at the facility as their main motivation for staying home. Even in these situations, however, the ultimate decision about where to seek deliver was made by household members independent of the respondent's wishes. These results highlighted the social dynamics within the household as an important site of intervention to further address inequitable access and the particular vulnerability of women living in homes without a norm of care seeking.

MJK responded to quantitative and qualitative study results by developing programs to address the barriers identified and improve equity and access by addressing two areas: 1) Empowerment of the women in relationship with her household members, and 2) access and adherence to iron-folic acid supply and JSY subsidies. For example, MJK has adapted the World Health Organization's Thinking Health Programme (THP), a lay counsellor-delivered perinatal mental health program that uses principles of cognitive behavioral therapy to improve maternal wellbeing by focusing on three areas: the mother's personal health, her relationship with her child, and her relationship with the people around her [54]. To address access and adherence to iron-folic acid, in-home therapy sessions on the mother's personal health included information on basic aspects of maternal health and a reinforcement of healthy behaviors, including the importance of iron-folic acid supplementation and providing information on how to obtain supplements, JSY subsidies, and other available services.

As shown, mixed-methods LQAS generated contextually specific quantitative and qualitative data that MJK could respond to. The Thinking Healthy Programme (THP) offered MJK an evidence-based platform that could be adapted to address the household norm of care seeking by confronting various issues that may emerge within specific households, including gender bias and discussions about domestic responsibility [54]. In MJK's adapted version of THP, in-home screening for depression and anxiety was used to identify pregnant women who may be vulnerable due to their mental illness, lack of access to care, or household dynamics, and a psychosocial intervention that addressed the family in an attempt to promote behavior change that would improve access.

## Fidelity

Fidelity to the study design was maintained with a few exceptions including the process for initial contact with ASHAs, sampling, and conduct of the survey. Although LQAS parameters called for a sample of 16 women per *panchayat*, our actual samples ranged from 16 to 20 women resulting in 25 additional women interviewed over the needed 288. This variation was unplanned, and a number of factors led to uncertainty regarding when to stop surveying. First, in order to sample an entire *panchayat* in a single day the research team would often split up and contact ASHAs in separate villages simultaneously and survey those villages separately. In this scenario, if one village did not produce the number of responses called for by the sampling plan, it was impossible to increase the number for the village being simultaneously surveyed in a sequential manner as planned. This challenge was in part because communication between the research teams to coordinate sampling in real time by mobile phone was often difficult due to the connectivity issues in some rural areas, especially in villages close to the international border. Further, without the opportunity to conduct a preliminary analysis of the survey data to ensure responses were complete and that no data were missing, the team was unsure if the minimum 16 responses needed for the primary LQAS indicators (FBD and full ANC) were achieved. Because of the distance, teams were also not able to revisit villages, so if there was time and eligible households, the research team surveyed more women from their respective randomized list than necessary. In cases where more than 16 households were surveyed per *panchayat*, these extra data were included in the data analyses, effectively reducing misclassification errors for those *panchayats*. Additional randomly sampled women would also enhance the precision of the point estimates through cluster sampling and increase the number of qualitative interviews.

Other adaptations to the original plan reflected local contextual factors identified during the study to increase feasibility and acceptability. These decisions were predicated on local variables often not foreseeable until arrival at the *panchayat*, such as distances and road

conditions between villages, weather, mobile phone connectivity, availability of the ASHA or *angadwadi*, and their willingness to act as a local guide. For example, the process of contacting ASHAs and obtaining their birth registries varied slightly with each *panchayat* and village depending on what circumstances were encountered. As a result, there was variability on how they were contacted (prior phone calls, contact through an intermediary, and sometimes simply arriving at the village unannounced) and where they were met (usually health centers but often individual houses or village schools).

## Costs

Total expenses, including salaries, transport, meals, translations and transcription, and data analysis was less than $8,000 USD, which is approximately $26 USD per household surveyed (n = 313).

## Discussion

We found that mixed-methods LC-LQAS as a novel methodology had high local acceptability of sampling procedures and question types, feasibility in the setting and within time and resource constraints, appropriateness for the study objectives, and effectiveness in terms of producing data needed to increase understanding of vulnerable women's experiences in accessing health care during pregnancy within the context of JSY and the National Rural Health Mission. Learning in the field and local constraints informed adaptations that affected some elements of implementation fidelity to the original plan but improved feasibility and did not affect core data collection and analysis. Costs were kept low despite the large amount of geographic area covered and remoteness of some supervision areas. Though the comparison is imperfect, mixed methods LC-LQAS was cheaper per household surveyed compared to the NFHS-4, with the caveat that NFHS-4 women's survey was longer (with 981 questions), took more time to complete (over 60 minutes in Rajasthan), and had different procedures [55]. The budget for the NFHS-4 was reportedly $33 million USD, or $55 USD per household surveyed (n = 601,509), compared to $25 USD per household in our study [56, 57].

Despite the relatively small number of women sampled over a geographic area of 3059 square kilometers, we were able to gather valuable qualitative data using questions that were specific enough to the respondent's circumstances but broad enough to elicit data that could be inductively coded and analyzed. We also arrived at theme saturation well before we coded 130 (60%) of the 215 audio-recorded short interviews. Future implementation of mixed-methods LC-LQAS must consider the tradeoff in time and resources required to apply rigorous research methods to analyze qualitative data from the embedded questions, which in our case took several months using a single coder (AB) or if more rapid methodologies are appropriate to the goals of the survey.

The targeted focus—pregnancy and delivery—facilitated extracting vivid and detailed narratives surrounding the seminal events in pregnancy and childbirth within the 20 to 40-minute window in which each survey was conducted. Based on the saturation, future use of mixed-methods LC-LQAS in this focus area could consider a subsampling of women for the embedded qualitative questions, which in our case was achieved at 41% of the surveyed household.

Measured against the objectives of the study, mixed-methods LC-LQAS demonstrated value in terms of 1) rapidly assessing FBD, ANC and JSY coverage for the population of interest at a local and subnational level, 2) uncovering aspects of lived experience that cannot emerge from the coverage statistics alone, and 3) providing actionable data to inform potential programming areas to improve health equity and access. The study was implemented fairly rapidly, at low cost, and was acceptable to both the target population and other local

stakeholders, including the local government. Among respondents, strategically embedding qualitative questions within the survey was both acceptable and effective at eliciting narrative data that uncovered the lived experience of pregnancy and childbirth. Key to the acceptability was the early engagement and approval from the local government, inclusion of ASHAs and *angadwadis* in the sampling process, and use of surveyors who were fluent in the local languages and culture and ready to adapt key processes such as how and where to contact the ASHAs and *angadwadis*. With few exceptions, ASHAs and *angadwadis* willingly granted access to official registers of pregnant women and voluntarily provided assistance in locating households, confirming eligibility, and facilitating access through introductions.

Mixed-methods LC-LQAS provided a useful tool for local health care providers working with vulnerable women in low-resource settings with a way to ground the statistical picture offered by population level surveys and provide the qualitative depth necessary to uncover and address inequity. For MJK staff, the importance of household norms around care access would not have been completely understood without this qualitative data. Conversely, confirming coverage data for our target population helped us contextualize and understand the qualitative findings. Mixed-methods LC-LQAS is therefore relevant for the shifting paradigm in health care towards an emphasis on people-centered, universal care that focuses on quality of health care delivery and patient experience. The World Health Organization's framework on Integrated People-Centered Health Services (IPCHS) calls for designing health systems "for people, with people" rather than around diseases and health institutions, which will require rigorous methods to engage communities and patients directly [20, 21]. This 'people-centered' approach is echoed in the Lancet Global Health Commission on High Quality Health Systems, which calls for care that is responsive to changing population needs, higher expectations, and ambitious health goals in the context of improvements in health outcomes in low-and-middle-income countries (LMICs) [58].

Our study and implementation of the survey had a number of limitations. The data did not allow for quantitative exploration of factors associated with FBD and ANC, which may have been useful for local managers. Since LC-LQAS is technically a multistage random sample, we could have completed such analysis; however, in this particular case the rare outcomes for ANC and its components would prevent having sufficient power to identify relevant predictors. However, this limitation also highlights one of the most unique aspects of our approach whereby we supplemented the quantitative data with qualitative interviews to draw out these facilitators and barriers.

While our methodology was able to rapidly provide rigorous and actionable qualitative and quantitative data, our ability to identify vulnerable women may have been limited. Vulnerability is a complex, socially determined status that we have assumed is synonymous with low socioeconomic status, which in turn we have assumed is indicated by the building material of the respondents' residence, which may have resulted in misclassification (for example, low-income or vulnerable women in *pucca* houses). It is recognized that categories such as 'scheduled caste', 'other backwards class', and BPL status, which in our sample represented only 19.3% of households (Table 3), are not accurate indicators of either poverty or vulnerability in India [59].

We likely missed relevant information by excluding women whose most recent pregnancy resulted in a miscarriage, stillbirth or infant death, although these represented less than 1% of women approached for an interview and were purposely avoided to reduce the risk of psychological harm. We were not able to capture the pregnancy experience of women who died during pregnancy or childbirth. Finally, we may have also introduced systematic bias by missing interviews with potentially eligible women who were away working or at their maternal village during pregnancy.

## Conclusion

We were able to successfully implement a novel mixed-method LC-LQAS survey to rapidly assess whether aggregate district-level estimates apply to a specific sub-population and understand not just what happened in terms of access, but also to uncover the lived experience of accessing reproductive care. The survey provided a way to understand why and how women were or were not successful in accessing care and provided the actionable knowledge needed by program managers to improve equity and impact. This ability to assess both the health care coverage and qualitative experience of a specific target population is an important feature of the method that addresses the pressing need to focus more on the experiences of vulnerable populations in local settings who, in the context of overall improvements in health access and outcomes, continue to fall through the cracks of the system. Understanding and addressing this inequity of access will help achieve the goal of Universal Health Care for all [59].

## Supporting information

**S1 File. LQAS parameters: Sample size, decision rule, coverage thresholds, and acceptable error.**
(DOCX)

**S2 File. LC-LQAS sample size calculation.**
(DOCX)

**S3 File. Minimal data set.**
(XLSX)

## Acknowledgments

This work was conducted as part of A.S.B.'s thesis project for the Master of Medical Sciences in Global Health Delivery program at Harvard Medical School. The authors acknowledge the substantial conceptual and methodological contributions made by Mary-Jo DelVecchio Good and Arlene M. Katz who, in addition to LRH and BH, were supervisors on this project. At Harvard, conceptual advice and support was also provided by Christina Lively, Joia Mukherjee, Paul Farmer, and Hannah Gilbert. The authors are grateful for the support of the staff and leadership at the Mata Jai Kaur Maternal and Child Health Centre. Data collection was conducted by a local research team and assisted by several Ashas and Angadwadis from district Sri Ganganagar. Further field research support was provided by Mantra4Change, Bengaluru and a student volunteer from Harvard. We thank our respondents for generously sharing their time and stories.

## Author Contributions

**Conceptualization:** Aneel Singh Brar, Bethany L. Hedt-Gauthier, Lisa R. Hirschhorn.

**Formal analysis:** Aneel Singh Brar.

**Methodology:** Aneel Singh Brar, Bethany L. Hedt-Gauthier, Lisa R. Hirschhorn.

**Supervision:** Bethany L. Hedt-Gauthier, Lisa R. Hirschhorn.

**Writing – original draft:** Aneel Singh Brar, Bethany L. Hedt-Gauthier, Lisa R. Hirschhorn.

**Writing – review & editing:** Aneel Singh Brar, Bethany L. Hedt-Gauthier, Lisa R. Hirschhorn.

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
