## [Decision Letter · Decision Letter 0]

12 Jan 2021

PONE-D-20-19007

Mixed Methods Lot Quality Assurance Sampling: A novel, rapid methodology to inform equity focused maternal health programming in rural Rajasthan, India

PLOS ONE

Dear Dr. Brar,

Thank you for submitting your manuscript to PLOS ONE. After careful consideration, we feel that it has merit but does not fully meet PLOS ONE’s publication criteria as it currently stands. Therefore, we invite you to submit a revised version of the manuscript that addresses the points raised during the review process.

The authors are advised to attend to the comments from both reviewers especially the second reviewer. Addressing these comments are important as anticipatory for peer readers (many are quite common for a manuscript of this type presented to quantitative methods oriented audience) and in assisting the consideration for publication. The manuscript is very well done.

We look forward to receiving your revised manuscript.

Kind regards,

Joseph Telfair, DrPH, MSW, MPH

Academic Editor

PLOS ONE

Journal Requirements:

2. We note that Figures 1 and 2 in your submission contain map images which may be copyrighted. All PLOS content is published under the Creative Commons Attribution License (CC BY 4.0), which means that the manuscript, images, and Supporting Information files will be freely available online, and any third party is permitted to access, download, copy, distribute, and use these materials in any way, even commercially, with proper attribution. For these reasons, we cannot publish previously copyrighted maps or satellite images created using proprietary data, such as Google software (Google Maps, Street View, and Earth). For more information, see our copyright guidelines: http://journals.plos.org/plosone/s/licenses-and-copyright.

You may seek permission from the original copyright holder of Figures 1 and 2 to publish the content specifically under the CC BY 4.0 license. 

If you are unable to obtain permission from the original copyright holder to publish these figures under the CC BY 4.0 license or if the copyright holder’s requirements are incompatible with the CC BY 4.0 license, please either i) remove the figure or ii) supply a replacement figure that complies with the CC BY 4.0 license. Please check copyright information on all replacement figures and update the figure caption with source information. If applicable, please specify in the figure caption text when a figure is similar but not identical to the original image and is therefore for illustrative purposes only.

Reviewers' comments:

Reviewer's Responses to Questions

**Comments to the Author**

1. Is the manuscript technically sound, and do the data support the conclusions?

Reviewer #1: Yes

Reviewer #2: Partly

2. Has the statistical analysis been performed appropriately and rigorously? 

Reviewer #1: Yes

Reviewer #2: Yes

3. Have the authors made all data underlying the findings in their manuscript fully available?

Reviewer #1: Yes

Reviewer #2: Yes

4. Is the manuscript presented in an intelligible fashion and written in standard English?

Reviewer #1: Yes

Reviewer #2: Yes

5. Review Comments to the Author

Reviewer #1: Thank you for the opportunity to review this manuscript which focuses on an innovative approach of Mixed Methods Lot Quality Assurance Sampling. This is a good manuscript that will make a significant contribution to literature around low cost sampling methods. The innovative approach tested in this study can provide a low-cost sampling design and data collection approach which will facilitate data driven decision making for local agencies working with limited funding.

There is scope for strengthening the manuscripts by adding a bit more explanation.

• Line 61: “This high burden has persisted in the face of great progress in reducing maternal and neonatal mortality with reductions of 68.7% and 50.9%, respectively.” What is the timeframe for this reduction? This change happened over X years?

• Line 179: “Given the most recent population size estimates for women aged 18-49 in Sri Ganganagar…”

While the reason for limiting these estimates to 18-49 years old women (instead of 15-49 years) is included later in the text, it will improve readability if the reason is included here.

• Line 272: Does CHW stand for community health worker?

• Line 273: “The perceived fit and relevance of mixed-methods LC-LQAS to achieve our study objectives within the given context and resource constraints.”

“Perceived” by whom? Researchers, government, or different stakeholders?

• Line 311: Table 2:

• Some content is in bold letters while others are not. What does bold statement signify? Maybe a footnote will be helpful.

• Line 315: “We captured refusal information from 14 of 18 panchayats…”

Is there a reason for excluding 2 villages from feasibility analysis? Are there any implications of leaving them out?

• Line 382: Table 3:

Including definition of demographic variables in the methods section will be helpful for readers from different country context to understand the table.

The row for age has “Mean USD, SD, 95%CI”. I think this should be “Mean years, SD, 95%CI”.

• Line 465: Add citation for Thinking Health Programme.

• Line 485: “Although LQAS parameters called for a sample of 16 women per panchayat, our actual samples ranged from 16 to 20 women resulting in 25 additional women interviewed over the needed 288”

Is there any implication of this oversampling?

• Line 515: Cost

Citing cost of other survey methods, if available, will be helpful to put this cost in perspective and understand how low the current cost is in comparison to other widely used methods.

Overall this manuscript is very well written. However, there are some grammatical and syntax error. I recommend the authors to proofread the manuscript.

Reviewer #2: I will focus on methods and reporting. The paper is well written and clear, in general.

Major

1) the LQAS power calculation is conducted correctly, but the choice of the approach and the resulting small sample affects the robustness of the research. It's an approach used for quality control, not large national health interventions at the country level, and that is a serious problem for the work submitted.

2) Unsurprisingly, the resulting sample is too small to inform the quantitative elements, especially with a dichotmous and rare primary outcome.

3) Also because of the small sample size, the authors are limited in presenting prevalence estimates for the outcomes of interest, rather than explore predictors, as should normally happen in such a design.

Minor

1) The abstract is long with no information on the methods used, qualitative or quantitative. Try to structure to improve.

6. PLOS authors have the option to publish the peer review history of their article (what does this mean?). If published, this will include your full peer review and any attached files.

Reviewer #1: No

Reviewer #2: No

---

## [Author Response · Author response to Decision Letter 0]

9 Mar 2021

Response to Editors 

Further clarification requested on February 19, 2021:

In response to the email on February 19, 2021 and subsequent correspondence with Agnes Pap, I have uploaded a new version of Figures 2 that was created by me using geospatial census data from the 2011 Indian Census provided by the Harvard Geospatial Library. Both figures 1 and 2 are now in compliance with copyright regulations. I made minor additional edits to the manuscript to accurately reflect the new figure 2.

Previously submitted clarification requested on February 3, 2021:

1) In the Methods, please clarify that participants provided oral consent. Please also state in the Methods:

- Why written consent could not be obtained

- Whether the Institutional Review Board (IRB) approved use of oral consent

- How oral consent was documented

2) You indicated that you had ethical approval for your study. In your Methods section, please ensure you have also stated whether you obtained consent from parents or guardians of the minors included in the study or whether the research ethics committee or IRB specifically waived the need for their consent.

Thank you for requesting that this important information is included. We have added the following information to the “Ethics” section of the Methodology, with new text in bold: 

The study was approved by the IRB at Harvard University Faculty of Medicine and the BioMedical Ethics Committee in New Delhi. Oral informed consent was obtained prior to the start of the survey. Oral consent was used for this population because of high levels of illiteracy, which is estimated at 67.2% for women in rural Sri Ganganagar, Rajasthan. If the participant was between the ages of 15 and 18, oral informed consent was obtained from their parent or guardian in addition to informed assent from the participant herself. Informed consent and assent were confirmed and documented on a form that was signed by the researcher.

3) Thank you for clarifying the sources of the data underlying Figures 1 and 2: a map image available from the Office of the Chief Electoral Officer Rajasthan

(www.ceorajasthan.nic.in) and open-source data (village-level polygons) available

from www.diva-gis.org.

Please confirm whether an image or data (or other content) from www.ceorajasthan.nic.in is actually used in your figure, or whether you simply used content from that site as a reference to fully build your own. This information will be helpful in determining whether further permissions are required at this time.

To further clarify, no part of that original map obtained from www.ceorajasthan.nic.in is used in Figures 1 or 2. The original map was only used as a reference to build our own images.

 

Response to Reviewer #1

Reviewer #1: Thank you for the opportunity to review this manuscript which focuses on an innovative approach of Mixed Methods Lot Quality Assurance Sampling. This is a good manuscript that will make a significant contribution to literature around low cost sampling methods. The innovative approach tested in this study can provide a low-cost sampling design and data collection approach which will facilitate data driven decision making for local agencies working with limited funding.

Thank you for taking the time to review our manuscript and for finding in our work the potential to make a significant contribution to low-cost sampling and data driven decision making in health care. We appreciate your detailed feedback and have addressed each point made below to hopefully strengthen the manuscript further. 

There is scope for strengthening the manuscripts by adding a bit more explanation.

• Line 61: “This high burden has persisted in the face of great progress in reducing maternal and neonatal mortality with reductions of 68.7% and 50.9%, respectively.” What is the timeframe for this reduction? This change happened over X years? 

You were correct in noting that improvement in mortality is not meaningful without a timeframe. We added the words “over a period of fifteen years” which was the period described in the maternal and child mortality trend reports cited. The first sentences of the article now read as follows with added text in bold:

In 2015, India accounted for 15% of all maternal deaths and 24% of all neonatal deaths globally (MMEIG, 2015; UN IGME, 2015). This high burden has persisted in the face of great progress in reducing maternal and neonatal mortality with reductions of 68.7% and 50.9%, respectively, over a period of fifteen years (MMEIG, 2015; UN IGME, 2015).

• Line 179: “Given the most recent population size estimates for women aged 18-49 in Sri Ganganagar…” While the reason for limiting these estimates to 18-49 years old women (instead of 15-49 years) is included later in the text, it will improve readability if the reason is included here. 

We agree with your assessment that the explanation should come earlier in the text. We moved the explanatory phrase from the “Study population and Sampling Procedures” subsection up to the “Large-Country Lot Quality Assurance Sampling: Indicators, Parameters and Sample Size” subsection to provide the rationale for using only women 18-49 years of age. The paragraph in question now reads as follows with added text in bold:

Given the most recent population size estimates for women aged 18-49 in Sri Ganganagar, the required sample size for randomly sampled clusters (panchayats) was 18, with 16 women per cluster, resulting in a total of 288 women. These calculations were performed on the basis of the number of women aged 18-49 years in each village from the 2011 Indian census (data on ages 15-17 years were not available) (S2 Supporting Information).

• Line 272: Does CHW stand for community health worker? 

Yes, thank you. We have added in the acronym in brackets after the first use of “community health worker”. This occurs in the “Assessment of the mixed-methods LC-LQAS” subsection under the definition for the “Acceptability” variable:

Acceptability: 1) The acceptance of the sampling procedures and surveys among respondents and their household members, community health workers (CHW), and local health bureaucrats; and 2) The acceptance of respondents to the integrated qualitative questions.

• Line 273: “The perceived fit and relevance of mixed-methods LC-LQAS to achieve our study objectives within the given context and resource constraints.” “Perceived” by whom? Researchers, government, or different stakeholders? 

Thank you. This is an important point in assessing the value of the methodology. We had defined “Acceptability” for this method as the perception of respondents and community members. We have added in similar details for “Appropriateness”, which is from the perspective of the program implementers, evaluators, researchers or other stakeholders using mixed-methods LC-LQAS to understand the success and challenges of program implementation or policy. We added the following bolded sentences to the “Appropriateness” definition:

Appropriateness: The perceived fit and relevance of mixed-methods LC-LQAS to achieve our study objectives within the given context and resource constraints. Appropriateness is from the perspective of program implementers, evaluators, researchers, or local stakeholders interested in understanding and improving implementation or policy.

• Line 311: Table 2: Some content is in bold letters while others are not. What does bold statement signify? Maybe a footnote will be helpful. 

Thank you for noting this. The bolding was originally meant to highlight predominant themes. We realised that this was confusing and so removed the bolding in Table 2.

• Line 315: “We captured refusal information from 14 of 18 panchayats…”

Is there a reason for excluding 2 villages from feasibility analysis? Are there any implications of leaving them out?

Apologies for the lack of clarity. We are addressing this comment with several additions throughout the text which strengthen the overall assessment of the method. We collected data on all households we approached (456 eligible households in total) for interviews in all 18 panchayats. However, we did not collect specific reasons not being able to complete the interviews (not at home, refusal as examples) for the first 4 panchayats before recognizing that this was an important implementation outcome to collect (“Acceptability”). For this reason, we only report on the panchayats for which we have refusal information in the results section. The clarify this point, we added the following sentences in bold to the “Acceptability” outcome definition in the “Assessment of the mixed-methods LC-LQAS” subsection of the Methods:

Acceptability: 1) The acceptance of the sampling procedures and surveys among respondents and their household members, community health workers (CHW), and local health bureaucrats; and 2) The acceptance of respondents to the integrated qualitative questions. To assess acceptability, we collected refusal information from eligible households that were approached for the survey. We recognized the importance of collecting refusal information after completing the first four panchayats, and so only collected and included this information from the subsequent 14.

Further, in the “Acceptability” sub-section of the Results, we added the following bolded phrase to the first paragraph:

In all 18 panchayats, we approached a total of 456 eligible households from which 313 (68.6%) women were surveyed, all of whom participated in both qualitative and quantitative components of the study, providing interviews of approximately 15-20 minutes in length. Five of these respondents (1.6%) refused audio recording.

We captured specific refusal information from 14 of 18 panchayats in which we approached 415 eligible households with 242 (48.3%) having a woman who delivered in the previous 2 years of the survey at home. Only three of the 242 (1%) refused to participate (1 language barrier, 1 refusal by head of household and 1 did not give consent).

Adding the total number of households approached also helped clarify our “Feasibility” outcome in the Results. In that section, we edited out that we surveyed 313 women and inserted the number households approached as the true extent of the survey scope. We edited the first paragraph of this section to read:

Implementing the mixed-methods LC-LQAS was feasible. We approached 456 eligible households in 18 panchayats consisting of 151 villages. The survey was completed over 19 days between September 16, 2015 and October 10, 2015 with all targeted panchayats reached, sample sizes achieved, and data collected according to the original or adapted protocols (see Fidelity).

Note that the “Feasibility” section of the results has detailed information on why households did not result in an interview other than refusal (such as participant was not home or the house was not eligible due to building material).

• Line 382: Table 3: Including definition of demographic variables in the methods section will be helpful for readers from different country context to understand the table.

This is an excellent suggestion. We added variable definitions to “Data Collection” section of the Methods.

Quantitative data collected included general demographic information, and for the most recent pregnancy, ANC attendance and receipt of and adherence to folate and iron supplementation, receipt of the JSY subsidy, and whether the delivery was facility-based. Demographic variables included household type (kachha or semi-pucca), education level, caste designation (Scheduled Caste, Scheduled Tribe, or Other Backwards Class) and Below Poverty Line (BPL) status. These caste and tribe designations are officially recognized in the Indian Constitution for groups of people who are economically or socially disadvantaged. BPL households are assessed as economically disadvantaged by local government agencies and provided with a BPL card which entitles them to certain welfare schemes.

The row for age has “Mean USD, SD, 95%CI”. I think this should be “Mean years, SD, 95%CI”. 

Thank you for catching this. USD has been changed to “years” in Table 3.

• Line 465: Add citation for Thinking Health Programme.

Great suggestion. We have added the reference #55 of the revised manuscript which references for the WHO’s Thinking Healthy Programme manual. The reference is:

55. World Health Organization. Thinking Healthy: A Manual for Psychosocial Management of Perinatal Depression (WHO generic field-trial version 1.0). Geneva: WHO; 2015. 

• Line 485: “Although LQAS parameters called for a sample of 16 women per panchayat, our actual samples ranged from 16 to 20 women resulting in 25 additional women interviewed over the needed 288” Is there any implication of this oversampling? 

Thank you for this question. Sampling additional women per panchayat would change the decision rule in determining whether a given panchayat or supervision area achieves defined threshold, as indicated in table 1. Increasing the sample size would also effectively reduce the misclassification error for those panchayats. Regarding the regional point estimates, these were obtained using cluster sampling, with all women randomly selected. This process should enhance the precision of the regional estimates for FBD, ANC, and JSY, and we adjusted the weights as needed to account for this oversampling in some areas. We discuss the deviation from the originally planned method in the “Fidelity” section of the Results with new sentences bolded:

Because of the distance, teams were also not able to revisit villages, so if there was time and eligible households, the research team surveyed more women from their respective randomized list than necessary. In cases where more than 16 households were surveyed per panchayat, these extra data were included in the data analyses, effectively reducing misclassification errors for those panchayats. Additional randomly sampled women would also enhance the precision of the point estimates through cluster sampling and increase the number of qualitative interviews.

• Line 515: Cost. Citing cost of other survey methods, if available, will be helpful to put this cost in perspective and understand how low the current cost is in comparison to other widely used methods. 

Thank you for the suggestion. To give a sense of the cost compared to other surveys, we made an imperfect comparison to the previous Indian National Family and Health Survey (NFHS-4) on a per household basis. We changed the “Cost” section of the results to provide a cost per household surveyed rather than cost per day. The Cost section now has the following paragraph with new text in bold:

Total expenses, including salaries, transport, meals, translations and transcription, and data analysis was less than $8,000 USD, which is approximately $26 USD per household surveyed (n=313).

In the Discussion, we added sentences to make an indicative comparison to the cost of the NFHS-4, with caveats about the comparability of the two methods. We added the following bolded sentences to the end of the first paragraph of the discussion with additional references to the NFHS-4 survey and a report on the cost of the survey:

Costs were kept low despite the large amount of geographic area covered and remoteness of some supervision areas. Though the comparison is imperfect, mixed methods LC-LQAS was cheaper per household surveyed compared to the NFHS-4, with the caveat that NFHS-4 women’s survey was longer (with 981 questions), took more time to complete (over 60 minutes in Rajasthan), and had different procedures (Srinivasan & Mishra, 2020). The budget for the NFHS-4 was reportedly $33 million USD, or $55 USD per household surveyed (n=601,509), compared to $25 USD per household in our study (International Institute for Population Sciences, 2017; Karpagam, 2019). 

The following references were added:

56. Srinivasan K, Mishra R. Quality of Data in NFHS-4 Compared to Earlier Rounds. Econ Polit Wkly. 2020;55(6):40–5. 

57. International Institute for Population Sciences. National Family Health Survey (NFHS-4) 2015-16 India [Internet]. Mumbai; 2017. Available from: http://rchiips.org/NFHS/NFHS-4Reports/India.pdf

58. Karpagam S. What ails India’s bedrock health survey: Exploited field workers, badly designed questionnaires Many gaps need to be addressed in planning the fifth National Family Health Survey. Scroll.in [Internet]. 2019 Feb 20; Available from: https://scroll.in/pulse/910955/what-ails-indias-bedrock-health-survey-exploited-field-workers-badly-designed-questionnaires

Overall this manuscript is very well written. However, there are some grammatical and syntax error. I recommend the authors to proofread the manuscript.

Thank you for finding the errors. We have re-read and corrected several grammatical and syntax errors.

 

Response to Reviewer #2:

Reviewer #2: I will focus on methods and reporting. The paper is well written and clear, in general.

Thank you for taking the time to review our manuscript and for providing us with valuable feedback regarding the applicability of LC-LQAS, a point that we could have clarified further. We respond to all three of your major comments below and have added sentences to the manuscript to address your concerns.

Major

1) the LQAS power calculation is conducted correctly, but the choice of the approach and the resulting small sample affects the robustness of the research. It's an approach used for quality control, not large national health interventions at the country level, and that is a serious problem for the work submitted.

This comment, along with your following comments question the applicability of LQAS to large national interventions at the country level, with concerns that the sample size is too small to report on the primary outcomes, and that the study does not explore predictors in addition to outcomes. Overall, we agree with your concerns about the applicability of LQAS for many contexts, and do not advocate for its use in all contexts for all questions.

However, we do believe from our past professional experiences that there are contexts where LQAS is not only appropriate but is actually preferred over other sampling strategies. We explain below and present changes to the manuscript that we believe address your concerns.

First, our goal was to test a method which would provide local and rapid data for program development and improvement. Specifically, the study was conducted to inform health program managers at the local level, and to give information about whether or not national FBD and ANC coverage data were applicable to one specific catchment area in district Sri Ganganagar, Rajasthan. This is particularly of interest for our context because existing national level surveys are often out of date due to their infrequency, and because those surveys are not specific enough to our intervention population of interest. We used LQAS because it has demonstrated utility in monitoring outcomes of interest in a timely fashion. The LQAS “classification” portion of this sampling is only on the local level, and that was by design. 

A second goal was to estimate FBD and ANC coverage for MJK’s target population across our total catchment area. We engaged Large Country LQAS to ensure that we sampled enough small areas to estimate this proportion with the desired level of precision. This estimate is important as MJK monitors the progress of FBD and ANC in its catchment area over time. 

To clarify both of these points we added the following bolded sentence to the end of the second paragraph in the “Settings and context” section of the Methods: 

The gap between the district-wide progress in coverage and the needs and experiences of MJK patients motivated the development of mixed-methods LC-LQAS. Our objective was to gather the evidence required to drive MJK’s program adaptation and to inform government health policy to improve equity in access and care. LQAS could provide MJK managers with a rapid assessment of whether their target population was receiving FBD and ANC. Integrating cluster sampling and qualitative methods adds value by providing regional point estimates that can be compared to national-level survey data and further contextual details to help inform program design.

2) Unsurprisingly, the resulting sample is too small to inform the quantitative elements, especially with a dichotomous and rare primary outcome.

We respond to this comment combined with the next comment.

3) Also because of the small sample size, the authors are limited in presenting prevalence estimates for the outcomes of interest, rather than explore predictors, as should normally happen in such a design.

We absolutely agree and for that reason we did not use the quantitative data to draw out factors associated with FBD and ANC. Because LC-LQAS is a multistage cluster sample, technically we could have done this predictor analysis. However, as you rightly point out: 1) our outcome is rare and 2) we were not designed for such analyses. The most novel factor of our approach is that we aimed to elucidate this information from the qualitative aspect of our study; so, while we did not design for predictive analyses and we would have poor power in this case because of the relatively rare outcome for ANC and its components, we do have insight into associated factors. We acknowledge your concerns and emphasize our novel approach with this edit (in bold) to the limitations at the end of the Discussion section:

Our study and implementation of the survey had a number of limitations. The data did not allow for quantitative exploration of factors associated with FBD and ANC, which may have been useful for local managers. Since LC-LQAS is technically a multistage random sample, we could have completed such analysis; however, in this particular case the rare outcomes for ANC and its components would prevent having sufficient power to identify relevant predictors. However, this limitation also highlights one of the most unique aspects of our approach whereby we supplemented the quantitative data with qualitative interviews to draw out these facilitators and barriers.

Minor

1) The abstract is long with no information on the methods used, qualitative or quantitative. Try to structure to imp

Thank you for this suggestion. We cut down extraneous information and added in a brief description of the methods. The revised abstract is provided below with added sentences in bold:

India has experienced a significant increase in facility-based delivery (FBD) coverage and reduction in maternal mortality. Nevertheless, India continues to have high levels of maternal health inequity. Improving equity requires data collection methods that can produce a better contextual understanding of how vulnerable populations access and interact with the health care system at a local level. While large population-level surveys are valuable, they are resource intensive and often lack the contextual specificity and timeliness to be useful for local health programming. Qualitative methods can be resource intensive and may lack generalizability. We describe an innovative mixed-methods application of Large Country-Lot Quality Assurance Sampling (LC-LQAS) that provides local coverage data and qualitative insights for both FBD and antenatal care (ANC) in a low-cost and timely manner that is useful for health care providers working in specific contexts. LC-LQAS is a version of LQAS that combines LQAS for local level classification with multistage cluster sampling to obtain precise regional or national coverage estimates. We integrated qualitative questions to uncover mothers’ experiences accessing maternal health care in the rural district of Sri Ganganagar, Rajasthan, India. We interviewed 313 recently delivered, low-income women in 18 subdistricts. All respondents participated in both qualitative and quantitative components. All subdistricts were classified as having high FBD coverage with the upper threshold set at 85%, suggesting that improved coverage has extended to vulnerable women. However, only two subdistricts were classified as high ANC coverage with the upper threshold set at 40%. Qualitative data revealed a severe lack of agency among respondents and that household norms of care seeking influenced uptake of ANC and FBD. We additionally report on implementation outcomes (acceptability, feasibility, appropriateness, effectiveness, fidelity, and cost) and how study results informed the programs of a local health non-profit.

---

## [Decision Letter · Decision Letter 1]

1 Apr 2021

Mixed Methods Lot Quality Assurance Sampling: A novel, rapid methodology to inform equity focused maternal health programming in rural Rajasthan, India

PONE-D-20-19007R1

Dear Dr. Brar,

We’re pleased to inform you that your manuscript has been judged scientifically suitable for publication and will be formally accepted for publication once it meets all outstanding technical requirements.

Kind regards,

Joseph Telfair, DrPH, MSW, MPH

Academic Editor

PLOS ONE

Additional Editor Comments (optional):

Reviewers' comments:

Reviewer's Responses to Questions

**Comments to the Author**

1. If the authors have adequately addressed your comments raised in a previous round of review and you feel that this manuscript is now acceptable for publication, you may indicate that here to bypass the “Comments to the Author” section, enter your conflict of interest statement in the “Confidential to Editor” section, and submit your "Accept" recommendation.

Reviewer #1: All comments have been addressed

Reviewer #2: All comments have been addressed

2. Is the manuscript technically sound, and do the data support the conclusions?

Reviewer #1: Yes

Reviewer #2: Yes

3. Has the statistical analysis been performed appropriately and rigorously? 

Reviewer #1: Yes

Reviewer #2: Yes

4. Have the authors made all data underlying the findings in their manuscript fully available?

Reviewer #1: Yes

Reviewer #2: Yes

5. Is the manuscript presented in an intelligible fashion and written in standard English?

Reviewer #1: Yes

Reviewer #2: Yes

6. Review Comments to the Author

Reviewer #1: Thank you for revising the manuscript. The manuscript now reflects the robustness of the study. I think the study is highly relevant and would help researchers in adopting new data collection approach for their future research.

Reviewer #2: I am happy with the authors' responses and the resulting changes to the paper. The limitations are being acknowledged.

7. PLOS authors have the option to publish the peer review history of their article (what does this mean?). If published, this will include your full peer review and any attached files.

Reviewer #1: No

Reviewer #2: No

---

## [Editor Report · Acceptance letter]

12 Apr 2021

PONE-D-20-19007R1 

Mixed Methods Lot Quality Assurance Sampling: A novel, rapid methodology to inform equity focused maternal health programming in rural Rajasthan, India 

Dear Dr. Brar:

I'm pleased to inform you that your manuscript has been deemed suitable for publication in PLOS ONE. Congratulations! Your manuscript is now with our production department. 

Kind regards, 

on behalf of

Dr. Joseph Telfair 

Academic Editor

PLOS ONE